# High Expression of EZH2 Mediated by ncRNAs Correlates with Poor Prognosis and Tumor Immune Infiltration of Hepatocellular Carcinoma

**DOI:** 10.3390/genes13050876

**Published:** 2022-05-13

**Authors:** Zhitao Chen, Xin Lin, Zhenmiao Wan, Min Xiao, Chenchen Ding, Pengxia Wan, Qiyong Li, Shusen Zheng

**Affiliations:** 1Shulan (Hangzhou) Hospital Affiliated to Zhejiang Shuren University Shulan International Medical College, Hangzhou 310004, China; 21818114@zju.edu.cn (Z.C.); 21918379@zju.edu.cn (X.L.); wanzm002@163.com (Z.W.); min.xiao@shulan.com (M.X.); dingcc0705@163.com (C.D.); pengxia.wan@shulan.com (P.W.); 2School of Medicine, Zhejiang University, Hangzhou 310003, China; 3School of Medicine, Zhejiang Chinese Medical University Zhejiang Shuren College, Hangzhou 310003, China

**Keywords:** hepatocellular carcinoma, *EZH2*, ncRNA, immune cell infiltration, prognosis

## Abstract

Background: Hepatocellular carcinoma (HCC) is the predominant form of liver cancer and is accompanied by a complex regulatory network. Increasing evidence suggests that an abnormal gene expression of *EZH2* is associated with HCC progression. However, the molecular mechanism by which non-coding RNAs (ncRNAs) regulate *EZH2* remains elusive. Methods: The Cancer Genome Atlas (TCGA) and Genotype-Tissue Expression (GTEx) data were used to perform differential expression analysis and prognostic analysis. We used the Encyclopedia of RNA Interactomes (ENCORI) database to predict candidate miRNAs and lncRNAs that may bind to *EZH2*. Subsequently, the comprehensive analysis (including expression analysis, correlation analysis, and survival analysis) identified ncRNAs that contribute to *EZH2* overexpression. Results: *EZH2* was found to be upregulated in the majority of tumor types and associated with a poor prognosis. Hsa-miR-101-3p was identified as a target miRNA of *EZH2*. Additionally, SNHG6 and MALAT1 were identified as upstream lncRNAs of hsa-miR-101-3p. Meanwhile, correlation analysis revealed that *EZH2* expression was significantly associated with the infiltration of several immune cell types in HCC. Conclusion: SNHG6 or MALAT1/hsa-miR-101-3p/*EZH2* axis were identified as potential regulatory pathways in the progression of HCC.

## 1. Introduction

Hepatocellular carcinoma (HCC) is the most common type of primary liver cancer, accounting for 90% of primary liver cancer cases, and is the leading cause of cancer-associated death worldwide [1,2]. Infection by hepatitis B virus (HBV) and hepatitis C virus (HCV), as well as alcohol consumption, is the predominant causative agent of HCC development, although non-alcoholic steatohepatitis associated with metabolic syndrome or diabetes mellitus is becoming a more frequent risk factor in the West [3,4]. Currently, the main methods for treating HCC are surgery, chemotherapy, radiotherapy, and liver transplantation [5]. In recent years, despite remarkable progress in immunotherapy, such as PD1-targeted therapy, there have still been a considerable number of HCC patients who cannot benefit from immunotherapy, which may be related to the immunosuppressive environment of tumors, resulting in the primary cause of mortality [6,7]. Therefore, elucidating the molecular mechanisms, especially the immune-related cellular mechanism, underlying the pathogenesis of HCC is essential for the development of effective anti-cancer therapies.

Enhancer of Zeste Homolog 2 (*EZH2*) is a member of the Polycomb group (PcG) family that forms multimeric protein complexes; the PcG family is involved in maintaining the transcriptional repressive state of genes over successive cell generations and in controlling the progression of the cell cycle, and it participates in the maintenance of cell differentiation [8,9]. As *EZH2* regulates cell cycle progression, its dysregulation accelerates cell proliferation and prolongs cell survival, which may lead to carcinogenesis and cancer development [10]. An increasing number of studies have revealed that *EZH2* is abnormally expressed in numerous tumors, including liver cancer, gastric cancer, and breast cancer, and correlates with tumor progression, metastasis, and drug resistance in prostate cancer cells [11,12,13]. In recent years, an increasing number of studies have suggested that *EZH2* may be a novel molecule involved in HCC progression, as well as a potential prognostic biomarker and therapeutic target [14,15]. Liu et al. [16] suggested *EZH2*/miR-622/CXCR4 as a potential adverse prognostic factor and therapeutic target for HCC patients. Similarly, Bae et al. shared the same view that overexpression of *EZH2* was an independent biomarker for poor outcomes of HCC, and that *EZH2* may be used as a therapeutic target in patients with HCC [17]. However, comprehensive studies on the expression, prognosis, mutation, and biological mechanisms of *EZH2* are still lacking in HCC. In addition, the relevance of *EZH2* to tumor immune infiltration is uncertain in HCC.

In the present study, we first analyzed *EZH2* expression and its prognostic value in a series of TCGA clinical samples of human cancers. Next, the molecular mechanisms underlying *EZH2*-mediated oncogenesis effects were explored in HCC. Subsequently, the relationships between *EZH2* expression and tumor-related immune cell infiltration, immune checkpoint blockade, and the immunotherapy response were explored in HCC.

## 2. Materials and Methods

### 2.1. Omics Analysis of EZH2

In this study, we aimed to explore the oncogenic role and potential biological mechanism of human *EZH2* in HCC. Firstly, we obtained the chromosome location, number of exons, and other biological information from the “gene” and “protein” modules of the National Center for Biotechnology Information (NCBI, https://www.ncbi.nlm.nih.gov/, accessed on 22 October 2021), U.S. National Library of Medicine. Secondly, the protein structure and conserved domains were explored via the Uniprot database (https://www.uniprot.org/, accessed on 22 October 2021). Thirdly, conserved amino acid sequences encoded by *EZH2* and the phylogenetic tree of the *EZH2* family were explored via the Constraint-based Multiple Alignment Tool (https://www.ncbi.nlm.nih.gov/tools/cobalt/, accessed on 22 October 2021) in NCBI. Finally, the distribution of the *EZH2* protein was obtained from the Human Protein Atlas (HPA, https://www.proteinatlas.org/, accessed on 25 October 2021) database.

### 2.2. Expression Analysis of EZH2

To evaluate the expression of *EZH2*, TIMER [18] (https://cistrome.shinyapps.io/timer/, accessed on 26 October 2021) was utilized to compare the differential expression levels of *EZH2* between tumor and normal tissues in various tumor types, including bladder urothelial carcinoma (BLCA), breast invasive carcinoma (BRCA), cervical squamous cell carcinoma and endocervical adenocarcinoma (CESC), cholangiocarcinoma (CHOL), colon adenocarcinoma (COAD), esophageal carcinoma (ESCA), glioblastoma multiforme (GBM), head and neck squamous cell carcinoma (HNSC), kidney chromophobe (KICH), kidney renal clear cell carcinoma (KIRC), kidney renal papillary cell carcinoma (KIRP), liver hepatocellular carcinoma (LIHC), lung adenocarcinoma (LUAD), lung squamous cell carcinoma (LUSC), pancreatic adenocarcinoma (PAAD), pheochromocytoma and paraganglioma (PCPG), prostate adenocarcinoma (PRAD), rectum adenocarcinoma (READ), stomach adenocarcinoma (STAD), thyroid carcinoma (THCA), and uterine corpus endometrial carcinoma (UCEC). Subsequently, we used the GEPIA2 database [19] (http://gepia2.cancer-pku.cn/#index, accessed on 2 November 2021) to validate the mRNA expression of the *EZH2* gene with data from The Cancer Genome Atlas (TCGA) and the Genotype-Tissue Expression (GTEx) project. Two-tailed Student’s *t*-tests were used to analyze the data, and a difference was considered to be statistically significant when *p* < 0.01 and |log2 FC (fold change)| ≥ 1.

### 2.3. Prognostic Analysis of EZH2

We used the “Survival Map” module of GEPIA2 to obtain the overall survival (OS) and disease-free survival (DFS) significance map of *EZH2* across all TCGA tumors. HCC patients from TCGA datasets were classified into high-risk and low-risk subgroups according to high (50%) and low (50%) cutoff values. The log-rank test was used as the hypothesis test, and the Kaplan-Meier (K-M) curves were also obtained through the “Survival Analysis” module of GEPIA2. The Sangerbox (http://sangerbox.com/, accessed on 4 November 2021) database was used to investigate how *EZH2* expression influenced tumor prognosis, including disease-specific survival (DSS) and progression-free interval (PFI).

### 2.4. Prediction of Upstream miRNAs and lncRNAs of EZH2

The Encyclopedia of RNA Interactomes (ENCORI, https://starbase.sysu.edu.cn/index.php, accessed on 8 November 2021) database [20] is an open-source platform for exploring miRNA-target interactions. We used ENCORI to predict candidate miRNAs and lncRNAs that may bind to *EZH2* and corresponding miRNAs. The upstream-binding miRNAs of *EZH2* were selected based on the following criterion: present in at least five of the following databases, consisting of PITA, RNA22, miRmap, microT, miRanda, PicTar, and TargetScan. The “pan cancer” module of ENCORI was used to perform the miRNA differential expression and survival analysis, miRNA-target co-expression, and RNA-RNA co-expression analysis.

### 2.5. Immune Cell Infiltration, Chemotactic Activities, Immune Cell Biomarkers, and Immune Checkpoint Analysis of EZH2

The “SCNA” module of TIMER (https://cistrome.shinyapps.io/timer/, accessed on 12 November 2021) was used to provide the comparison of tumor infiltration levels among tumors with different somatic copy number alterations for the *EZH2* gene. In addition, the “Gene” module of TIMER was also used to visualize the correlation of *EZH2* expression with the immune infiltration level in HCC tissues. We assessed the relationships among *EZH2* expression and immune cell chemotaxis, immune cell biomarkers, and immune checkpoints based on the “Correlation” module of TIMER. GEPIA2 was used to verify the correlation between *EZH2* expression and immune cell biomarkers.

### 2.6. Functional Analysis of EZH2

To explore the potential biological function and pathway relationships of *EZH2*, gene set enrichment analysis (GSEA) was performed using the Sangerbox (http://sangerbox.com/, accessed on 22 November 2021) online service. The top terms of the Kyoto Encyclopedia of Genes and Genomes (KEGG) and HALLMARK analyses were exhibited.

## 3. Results

### 3.1. Omics Analysis of EZH2

The goal of this study was to investigate the oncogenic role of *EZH2* in HCC. *EZH2* (Gene ID: 2146) is encoded on chromosome 7q36.1 and contains twenty-five exons (Figure 1A). The secondary structure of the EZH2 protein sequence is shown (Figure 1A). The *EZH2* oncogenic gene encodes five main protein isoforms consisting of histone-lysine N-methyltransferase *EZH2* isoforms a–e, which are mainly distributed in the nucleoplasm (Figure 1B). To better understand the oncogenic role of *EZH2*, structure–function analysis was conducted, and the protein domains are displayed (Figure 1C). *EZH2* contains an *EZH2*_WD binding (pfam11616) domain and a SET (cl02566) domain and is highly conserved in multiple species (Figure 1E). The phylogenetic tree of the *EZH2* protein was produced using fast minimum evolution, and it presents the evolutionary relationships among different species (Figure 1D).

### 3.2. Pan-Cancer Analysis of EZH2 Expression

To explore the possible carcinogenic roles of *EZH2*, differential expression analyses were conducted in 21 types of human cancer. As shown in Figure 2A, the expression level of *EZH2* in tumor tissues was significantly higher than in the corresponding normal tissues, including BLCA, BRCA, CESC, CHOL, COAD, ESCA, GBM, HNSC, KIRC, KIRP, LIHC, LUAD, LUSC, PRAD, READ, STAD, THCA, and UCEC. However, no significant difference in *EZH2* in KICH, PAAD, and PCPG was observed. To further validate the expression levels of *EZH2* in 21 types of human cancer, the GEPIA2 database, including the TCGA and GTEx datasets, was employed. Compared with normal tissues, *EZH2* was significantly upregulated in 16 cancer types, including BLCA, BRCA, CESC, CHOL, COAD, GBM, HNSC, KIRC, KIRP, LIHC, LUAD, LUSC, PAAD, READ, STAD, and UCEC, but not in others (Figure 2B). These results demonstrate that upregulated *EZH2* can support tumor growth and further imply that it is a crucial regulator in carcinogenesis for 15 types of cancer, including BLCA, BRCA, CESC, CHOL, COAD, GBM, HNSC, KIRC, KIRP, LIHC, LUAD, LUSC, READ, STAD, and UCEC.

### 3.3. Prognostic Analysis of EZH2 in Human Cancers

To determine whether *EZH2* expression levels are correlated with the prognosis of cancer patients, we evaluated the prognostic value of *EZH2* in cancer using the GEPIA2 database. For these 15 cancers, four prognosis-related indicators, including OS, DFS, PFI, and DSS, were used to evaluate the prognostic value of *EZH2*. In OS analysis, high expression of *EZH2* predicted worse survival in patients with LIHC or KIRC (Figure 3A). DFS analysis showed that high *EZH2* expression is correlated with poor prognosis for the TCGA cases of BLCA, KIRP, and LIHC (Figure 3B). In PFI analysis, high expression of *EZH2* was found to serve as an indicator of worse prognosis in KICH, KIRC, KIRP, LIHC, STAD, and UCEC (Figure 4A–F). In addition, high expression of *EZH2* was also associated with a shorter DFI in patients with KIRC, KIRP, LIHC, STAD, STAD, and UCEC (Figure 4G–K). Through the combination of the four prognosis-related indicators, upregulated *EZH2* may be utilized as an unfavorable prognostic biomarker in patients with HCC.

### 3.4. Prediction of Upstream miRNAs of EZH2

Non-coding RNA (ncRNA) comprises RNA molecules that do not encode a protein but regulate gene expression at multiple levels, including RNA splicing, editing, chromatin structure, and transcription [21]. To determine whether *EZH2* was regulated by some ncRNAs, the ENCORI database was used to predict upstream miRNAs that could potentially bind to *EZH2*. As shown in Figure 5A, 12 miRNAs were identified, namely, hsa-miR-137, hsa-miR-217, hsa-miR-32-5p, hsa-miR-363-3p, hsa-miR-367-3p, hsa-miR-92a-3p, hsa-miR-92b-3p, hsa-miR-101-3p, hsa-miR-1297, hsa-miR-138-5p, hsa-miR-26a-5p, and hsa-miR-26b-5p, which might be involved in regulating the expression of the regulators by targeting *EZH2*. The results also show that hsa-miR-137 (R = 0.319, *p* = 3.36 × 10^−^^10^), hsa-miR-363-3p (R = 0.164, *p* = 1.52 × 10^−3^), hsa-miR-92b-3p (R = 0.271, *p* = 1.19 × 10^−7^), and hsa-miR-138-5p (R = 0.147, *p* = 4.57 × 10^−3^) positively regulated *EZH2* expression (Figure 5B). However, hsa-miR-101-3p (R = −0.328, *p* = 9.68 × 10^−11^) and hsa-miR-26b-5p (R = −0.114, *p* = 2.83 × 10^−2^) negatively regulated *EZH2* expression (Figure 5B). The general idea of the combined analysis of miRNA and mRNA is to find the target gene and target miRNA according to the negative correlation between miRNA expression and target gene expression given the targeted relationship between miRNA and mRNA [22]. Ultimately, hsa-miR-101-3p and hsa-miR-26b-5p were identified as the upstream miRNAs that could potentially bind to *EZH2*. We further examined whether hsa-miR-101-3p and hsa-miR-26b-5p were involved in the regulation of the expression of *EZH2* in HCC based on expression and survival analysis. The results demonstrate that hsa-miR-101-3p (*p* = 6.80 × 10^−31^) was markedly downregulated in HCC and its upregulation was positively linked to patients’ prognosis (HR = 0.57, *p* = 1.70 × 10^−3^) (Figure 5C,E). Furthermore, hsa-miR-26b-5p (*p* = 2.90 × 10^−22^) was markedly downregulated in HCC (Figure 5D). However, the result indicates that hsa-miR-26b-5p overexpression was not associated with LIHC prognosis (HR = 1.16, *p* = 0.41) (Figure 5F). Meanwhile, the pairing information of hsa-miR-101-3p and *EZH2* is displayed (Figure 5G). Taken together, these results indicate that hsa-miR-101-3p regulation of *EZH2* expression might be involved in LIHC progression.

### 3.5. Prediction of Upstream lncRNAs of hsa-miR-101-3p

Here, we explored upstream lncRNAs of hsa-miR-101-3p based on the ENCORI database. The results reveal that 63 possible lncRNAs were identified as upstream lncRNAs of hsa-miR-101-3p. We assessed these lncRNAs’ differential expression levels between tumor and normal tissues using data from the TCGA and GTEx databases. Among the selected lncRNAs, we only identified three lncRNAs, namely, AC239868.3, SNHG6, and MALAT1, whose expression was significantly upregulated in HCC compared with normal controls (Figure 6A–C). Additionally, the pairing information of hsa-miR-101-3p–SNHG6, hsa-miR-101-3p–MALAT1, and hsa-miR-101-3p–AC239868.3 is displayed (Figure 6D–F). Meanwhile, we found that different pathological stages of HCC showed higher SNHG6 or MALAT1 expression compared with normal tissues (Figure 6G,H). Unfortunately, none of the data could be used to assess the correlation of AC239868.3 between different pathological stages of HCC and normal tissues. According to the competing endogenous RNA (ceRNA) theory, lncRNAs can act as endogenous RNAs and thereby regulate target gene transcripts by competing with shared miRNAs [23]. Thus, the correlation of expression values between the miRNA and the lncRNAs must be negative, and the correlation values between the lncRNAs and the mRNA must be positive. Subsequently, pairwise correlations between mRNA, miRNAs, and lncRNAs were explored to identify collinearity using the ENCORI database (Figure 7A–F). Eventually, SNHG6 and MALAT1 were determined to be directly targeted by hsa-miR-101-3p, and AC239868.3 was ultimately not identified as a potential upstream lncRNA of hsa-miR-101-3p.

### 3.6. Immune Cell Infiltration and Chemotactic Activity Analysis of EZH2 in LIHC

Given the known role of immune cells in tumor pathogenesis and the involvement of *EZH2* in immune cell development, differentiation, and function [24], the identification of *EZH2* expression associated with tumor-related immune cell infiltration will facilitate the monitoring of the HCC immunotherapy response and the exploration of the immune infiltration mechanism. We investigated the relationship between *EZH2* and immune cell infiltration in HCC and found that the high copy number amplification in B cells, CD8+ T cells, macrophages, neutrophils, and dendritic cells indicated significantly downregulated expression in HCC (Figure 8A). Immune infiltration analysis revealed a significant correlation between the expression of *EZH2* and the abundance of immune cell infiltration, including B cells, CD4+ T cells, CD8+ T cells, macrophages, neutrophils, dendritic cells, and cancer-associated fibroblasts (CAFs), in LIHC tissues (Figure 8B,C). Many of these factors, including CXCL chemokines and CCL chemokines, are important for the regulation and chemotaxis of immune cells, especially monocytes/macrophages, T lymphocytes, and eosinophils [25]. As listed in Table 1, *EZH2* expression was significantly positively correlated with monocyte-/macrophage-related chemokines (CCL5, CCL7, CCL8, and CCL13), T lymphocyte-related chemokines (CCL1), mast cell-related chemokines (CCR1, CCR2, CCR3, CCR4, CCR5, CXCR2, and CXCR4), eosinophil-related chemokines (CCL26, CCL5, CCL13, and CCL5), and neutrophil-related chemokines (CXCL8). Collectively, these results are in line with expectations and strongly suggest that *EZH2* is positively linked to immune cell infiltration and chemotactic activities and plays a vital role in LIHC immunity.

### 3.7. Expression Correlation of EZH2 and Biomarkers of Immune Cells in HCC

To further validate the notion that *EZH2* is positively associated with immune cell infiltration in HCC, we investigated the relationship between *EZH2* expression and the representative immune markers of several immune cells, including B cells, CD8+ T cells, CD4+ T cells, M1 macrophages, M2 macrophages, neutrophils, and dendritic cells (Table 2). In the GEPIA2 database, the expression levels of *EZH2* were found to be strongly correlated with most immune markers, including B cells (CD19), CD8+ T cells (CD8A and CD8B), CD4+ T cells (CD4), M1 macrophages (IRF5), M2 macrophages (CD163, VSIG4, and MS4A4A), neutrophils (ITGAM), and dendritic cells (HLA-DPB1, HLA-DRA, HLA-DPA1, CD1C, NRP1, and ITGAX). Similar results were observed in the TIMER database. Taken together, the findings indicate that these immune marker genes play a key role in immune cell infiltration, indicating that *EZH2* may be involved in immune surveillance and immune escape.

### 3.8. Correlation between EZH2 Expression and Immune Checkpoints in HCC

Immunotherapy based on PD-1/PDL1 and CTLA-4 has emerged as a new pillar of cancer treatment for patients with HCC [26]. To determine the influence that *EZH2* expression had on immunotherapy in patients with LIHC, we next evaluated the relationship between *EZH2* expression and PD-1, PD-L1, or CTLA-4 based on two different databases. For TIMER, the expression of *EZH2* in HCC was significantly positively correlated with PD-1, PD-L1, and CTLA-4 (Figure 9A–C). We observed the same positive correlation in GEPIA (Figure 9D–F). Thus, these results imply that positive *EZH2* expression may predict a better response to immunotherapy than negative expression.

### 3.9. Functional Analysis of EZH2 by GSEA

GSEA was performed to explore the biological role of *EZH2*. The KEGG enrichment terms indicated that high expression of *EZH2* is mainly associated with the cell cycle, homologous recombination, and nucleotide excision repair, while low expression is mainly associated with asthma, complement and coagulation cascades, primary bile acid biosynthesis, and arachidonic acid metabolism (Figure 10A,B). HALLMARK terms indicated that high expression of *EZH2* is associated with the G2M checkpoint, E2F targets, and mtorc1 signaling, while low expression of *EZH2* is associated with the p53 pathway, myogenesis, bile acid metabolism, and coagulation (Figure 10C,D). These results suggest the possible signaling pathway and mechanism associated with EZH2’s role in immune and metabolic functioning.

## 4. Discussion

Liver cancer ranks sixth in terms of incidence among malignancies and is the fourth leading cause of tumor-related death worldwide [1]. HCC is a highly heterogeneous disease that has been documented at the interpatient, intertumoral, and intratumoral levels, which makes its effective treatment challenging [27,28]. Though some progress has been made in the treatment of HCC, such as surgical resection, microwave ablation, and liver transplantation, the prognosis of HCC patients remains poor. Therefore, exploring the pathogenesis of HCC and identifying new targets to combat HCC are urgently needed and possess great significance for its clinical treatment.

*EZH2* encodes a member of the PcG family, which is involved in maintaining the transcriptional repressive state of genes over successive cell generations [29]. Existing studies have recognized the critical roles played by *EZH2* in tumor angiogenesis and cell proliferation, as well as cell differentiation and apoptosis [30]. Some investigations have demonstrated ncRNA to be closely associated with the occurrence of HCC and its dysfunction to inhibit tumor growth and metastasis [31]. The importance of the immune status in the tumor microenvironment (TME) has been gradually recognized in recent years [32,33]. In HCC, the TME is immunosuppressive and promotes immune tolerance and evasion by various mechanisms, promoting tumor proliferation, invasion, and metastasis [34]. Indeed, either *EZH2* or ncRNA can regulate inflammation and participate in immune gene expression, thus affecting the TME [24,35]. Thus, to determine the factor that influences the immunosuppression of the TME and the clinical response of immunotherapy, we need to explore some immunological genes affecting the abundance of immune cells in the TME. Targeted research may significantly change the clinical outcome of HCC.

Increasing evidence has addressed the role of *EZH2* in different human malignancies, including ovarian cancer, pancreatic cancer, gastric cancer, and even HCC [36,37,38,39]. *EZH2* has been reported to promote the recurrence and progression of HCC and thus is an important factor for tumor growth [36,40]. Previous studies have revealed that high *EZH2* expression may represent a novel indicator of poor prognosis in patients with HCC [40,41]. These results are consistent with our present study. In the present study, pan-cancer expression and survival analyses were performed on *EZH2* using TCGA datasets, and we found that EZH2 was abnormally expressed in 15 types of cancer, including BLCA, BRCA, CESC, CHOL, COAD, GBM, HNSC, KIRC, KIRP, LIHC, LUAD, LUSC, READ, STAD, and UCEC. In particular, high *EZH2* expression in HCC tissues was associated with poor prognosis. The expression results were validated in GEPIA2 using the TCGA and GTEx datasets. Furthermore, we found that the *EZH2* expression levels in liver cancers at advanced clinicopathological stages were significantly higher than those in tumors at early stages, implying that increased *EZH2* expression may indicate tumor progression in these patients. These reports, together with our analytic results, show the oncogenic role of *EZH2* in HCC.

The published literature has largely focused on the role of these regulatory ncRNAs in cancer initiation and progression [42]. Firstly, to explore the upstream regulatory miRNAs of *EZH2*, we introduced seven prediction programs, namely, PITA, RNA22, miRmap, microT, miRanda, PicTar, and TargetScan, to predict possible miRNAs that could potentially bind to EZH2. At the end, twelve upstream miRNAs of *EZH2* were confirmed using bioinformatics database prediction. Subsequently, differential expression analysis and mRNA–miRNA correlation analysis were performed to determine hsa-miR-101-3p as the upstream miRNA of *EZH2* affecting the progression of patients with HCC. Next, upstream lncRNAs of the hsa-miR-101-3p/*EZH2* axis were also predicted, and 63 possible lncRNAs were found. By conducting expression analysis and correlation analysis, two of the most potential upregulated lncRNAs, namely, SNHG6 and MALAT1, were identified as the upstream lncRNAs of hsa-miR-101-3p. In our study, hsa-miR-101-3p was shown to be downregulated in HCC tissues compared to normal tissues, and low hsa-miR-101-3p indicated a poor prognosis for HCC patients. SNHG6 and MALAT1 also demonstrated differential expression between normal tissues and tumor tissues. The in vitro experiments confirmed that HBV downregulated hsa-miR-101-3p expression by inhibiting its promoter activity, which resulted in the upregulation of Rap1b, and the downregulation of hsa-miR-101-3p or upregulation of Rap1b promoted the proliferation and migration of HCC cells [43]. SNHG6 may act as a competing endogenous RNA, effectively becoming a sink for hsa-miR-101-3p and thereby modulating the de-repression of zinc finger E-box binding homeobox 1, imposing an additional level of post-transcriptional regulation [44]. Likewise, dysregulation of MALAT1 has been found to participate in HCC progression [45,46]. Taken together, SNHG6 or MALAT1/hsa-miR-101-3p/*EZH2* axis were identified as potential regulatory pathways in HCC (Figure 11).

Immunotherapies have emerged as promising therapeutic strategies in HCC. Tumor-infiltrating immune cells in the TME affect responsiveness to such therapies, as well as outcomes [47,48]. Thus, we further characterized the relationship between *EZH2* expression and the infiltration levels of tumor-infiltrating immune cells in HCC tissues. We found that *EZH2* expression was significantly positively correlated with various immune cells, including B cells, CD4+ T cells, CD8+ T cells, macrophages, neutrophils, dendritic cells, and CAFs, in HCC tissues. Meanwhile, *EZH2* showed a positive correlation with biomarkers of immune cells and the chemotactic activity of tumor-related immune cells. From these findings, we speculated that immune cell infiltration might partially account for *EZH2*-mediated oncogenic roles and participate in the proliferation, migration, and immune response in HCC. However, the function of *EZH2* and its roles in hepatocarcinogenesis and progression need to be explored through further clinical and experimental studies.

Currently, immune checkpoint inhibitors (ICB), such as anti-CTLA-4 and anti-PD-L1/PD-1 antibodies, elicit durable and effective responses in some solid tumors [49]. However, the efficacy and side effects for each patient during treatment show individual differences [50]. Therefore, it is necessary to identify patients who might benefit from ICB therapy. In our study, the expression level of *EZH2* was significantly positively correlated with PD-1, PD-L1, and CTLA-4, which provided potential immunotherapy targets and indicated a better response to the immune-inhibiting reagents in patients with high *EZH2* expression. The underlying mechanism of the relationship between *EZH2* and ICB requires further exploration.

## 5. Conclusions

In summary, we found that EZH2 was highly expressed in multiple types of human cancer (including HCC) and was associated with a poor prognosis in HCC. We constructed an ncRNA-mediated regulatory mechanism of EZH2 in hepatocarcinogenesis and progression, namely, EZH2-hsa-miR-101-3p-SNHG6/MALAT1. Additionally, we found that EZH2 expression was not only associated with immune cell infiltration but also correlated with the expression of immune checkpoint genes.

## Figures and Tables

**Figure 1 genes-13-00876-f001:**
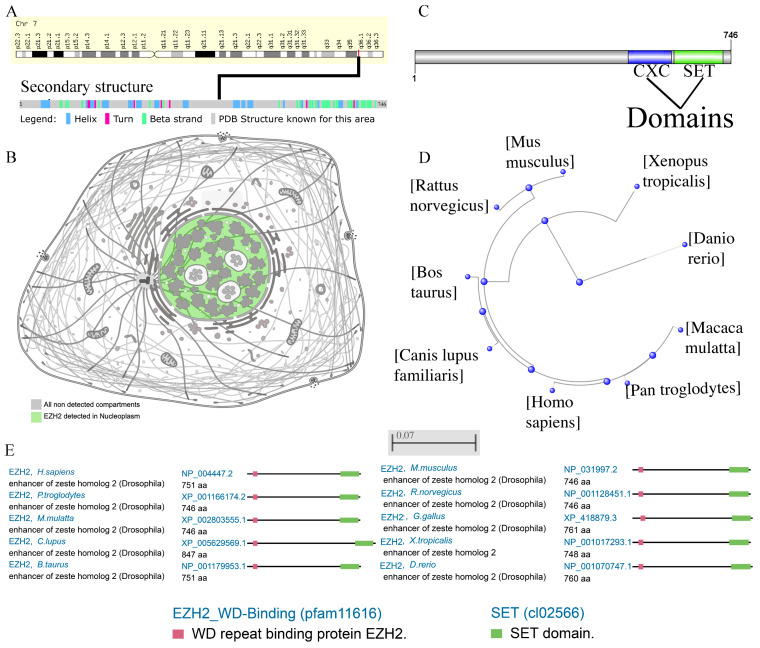
Chromosome localization, protein localization, and conservation analysis of *EZH2*. (**A**) Chromosome localization and protein secondary structure of *EZH2* in humans. (**B**) The main location of the *EZH2* protein in cells. (**C**) The conserved domain of *EZH2* in the amino acid sequence. (**D**) The phylogenetic tree of *EZH2* in different species. (**E**) Conservation of the *EZH2* protein among different species.

**Figure 2 genes-13-00876-f002:**
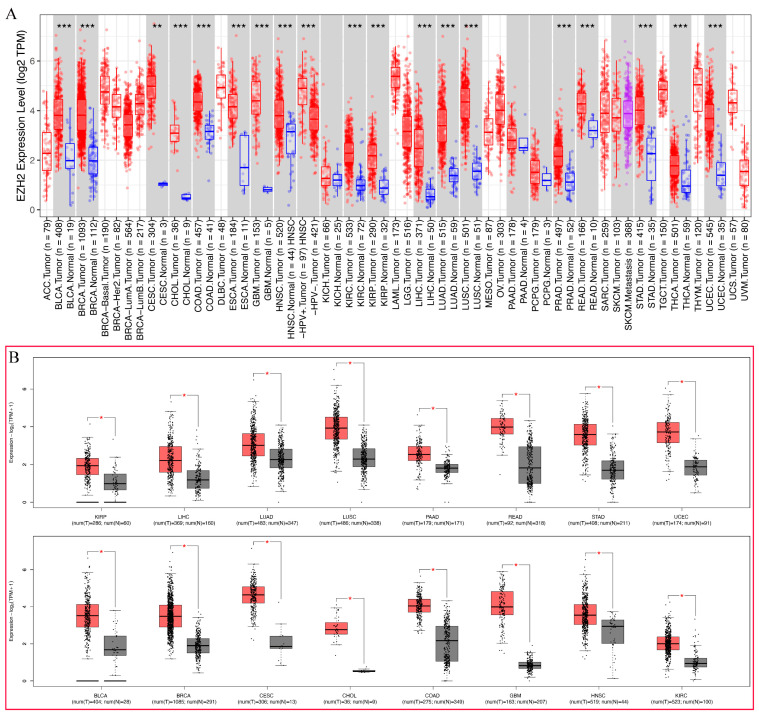
The differential expression of *EZH2* in different tumor types. (**A**) The expression status of the *EZH2* gene in different cancer types was analyzed through TIMER2 (data from TCGA). ** *p* < 0.01; *** *p* < 0.001. (**B**) The expression status of the *EZH2* gene in different cancer types was analyzed through GEPIA2 (data from TCGA and GTEx). * *p* < 0.05.

**Figure 3 genes-13-00876-f003:**
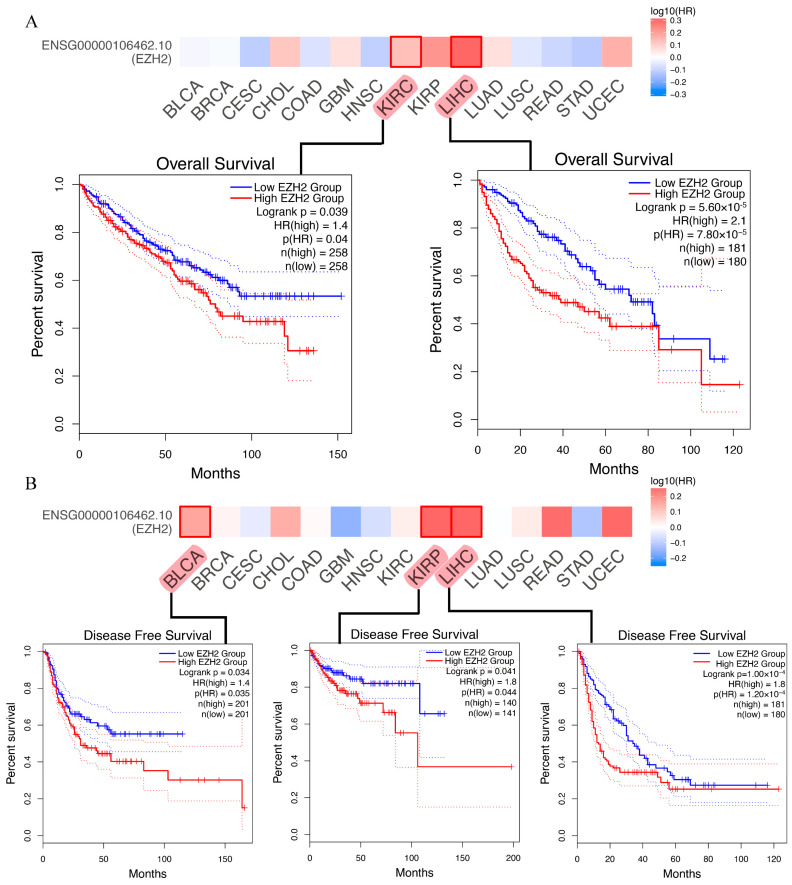
Kaplan-Meier survival curves of human cancers with high and low *EZH2* expression analyzed by the GEPIA2 and Sangerbox databases. (**A**) Overall survival (OS). (**B**) Disease-free survival (DFS).

**Figure 4 genes-13-00876-f004:**
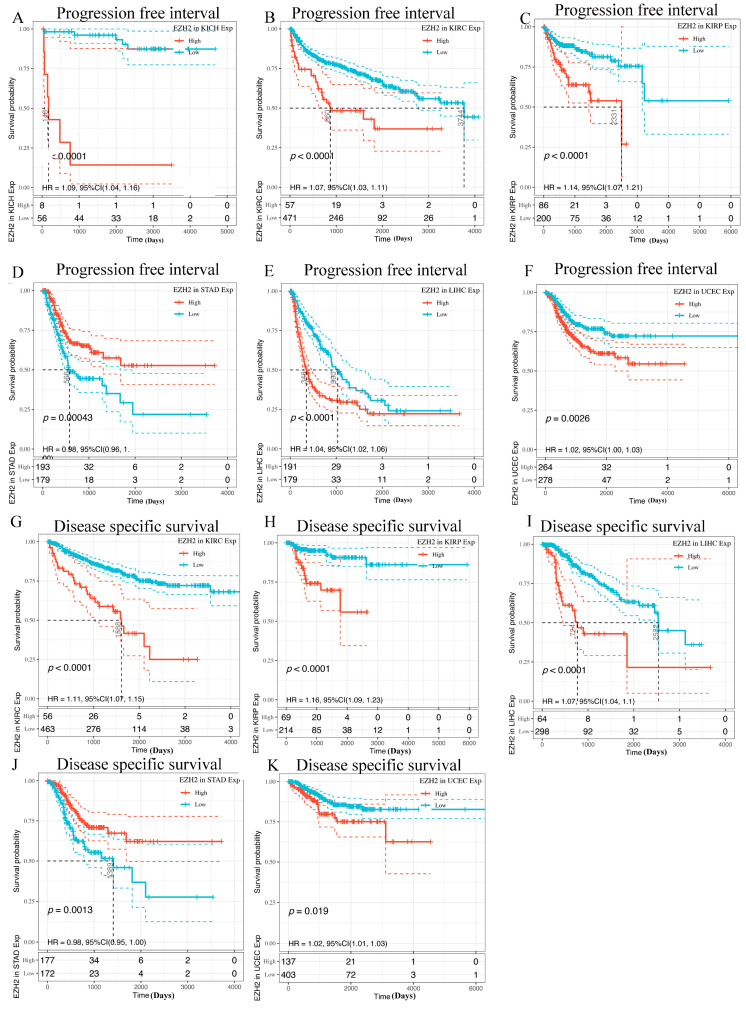
Kaplan-Meier survival curves of human cancers with high and low *EZH2* expression analyzed by the GEPIA2 and Sangerbox databases. (**A**–**F**) Progression-free interval (PFI). (**G**–**K**) Disease-specific survival (DSS).

**Figure 5 genes-13-00876-f005:**
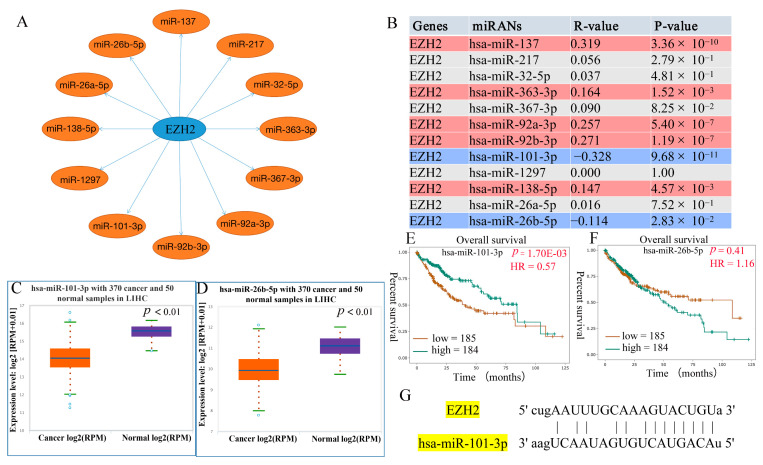
Prediction and identification the potential upstream regulatory miRNAs of *EZH2* in HCC. (**A**) The predicted miRNA-*EZH2* network. (**B**) The correlation between the candidate miRNAs and *EZH2* in HCC, where red represents a positive correlation and blue represents a negative correlation. (**C**,**D**) Differential expression analysis of hsa-miR-101-3p and hsa-miR-26b-5p in HCC tissues and normal tissues. (**E**,**F**) Prognostic analysis of hsa-miR-101-3p and hsa-miR-26b-5p in HCC. (**G**) Pairing information of hsa-miR-101-3p and *EZH2*.

**Figure 6 genes-13-00876-f006:**
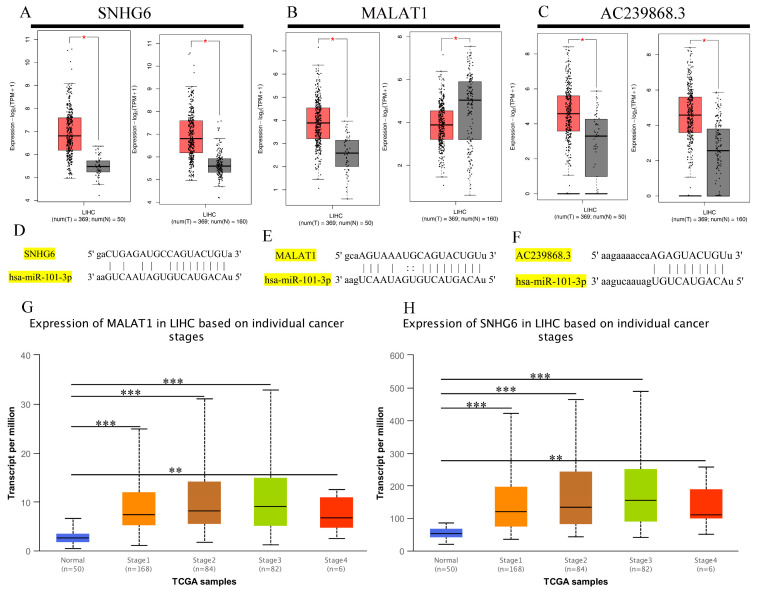
The differential expression level of upstream lncRNAs of hsa-miR-101-3p in different tumors and pathological stages. (**A**–**C**) The expression of SNHG6, MALAT1, and AC239868.3 in TCGA HCC compared with “TCGA and (or) GTEx normal” data. (**D**–**F**) Pairing information of hsa-miR-101-3p–SNHG6, hsa-miR-101-3p–MALAT1, and hsa-miR-101-3p–AC239868.3. (**G**,**H**) SNHG6 and MALAT1 differential expression in HCC with individual cancer stages. * *p* < 0.05; ** *p* < 0.01; *** *p* < 0.001.

**Figure 7 genes-13-00876-f007:**
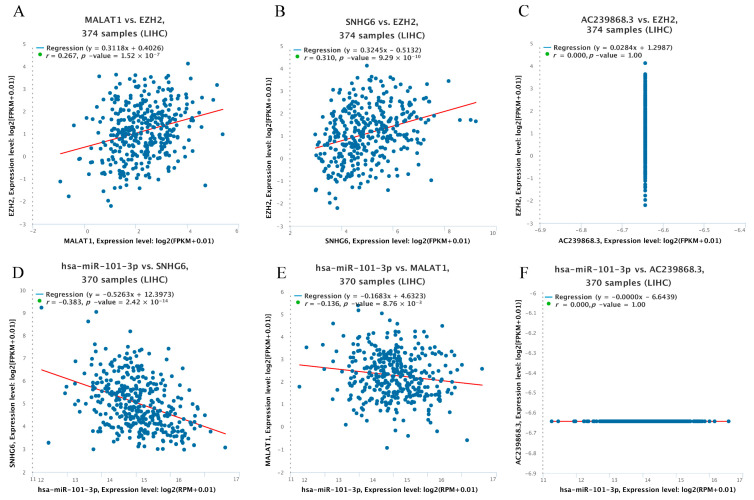
Correlation analysis between lncRNAs and hsa-miR-101-3p or between lncRNAs and EZH2 in HCC. (**A**) SNHG6 and EZH2, (**B**) MALA1 and EZH2, (**C**) AC239868.3 and EZH2, (**D**) hsa-miR-101-3p and SNHG6, (**E**) hsa-miR-101-3p and MALA1, and (**F**) hsa-miR-101-3p and AC239868.3.

**Figure 8 genes-13-00876-f008:**
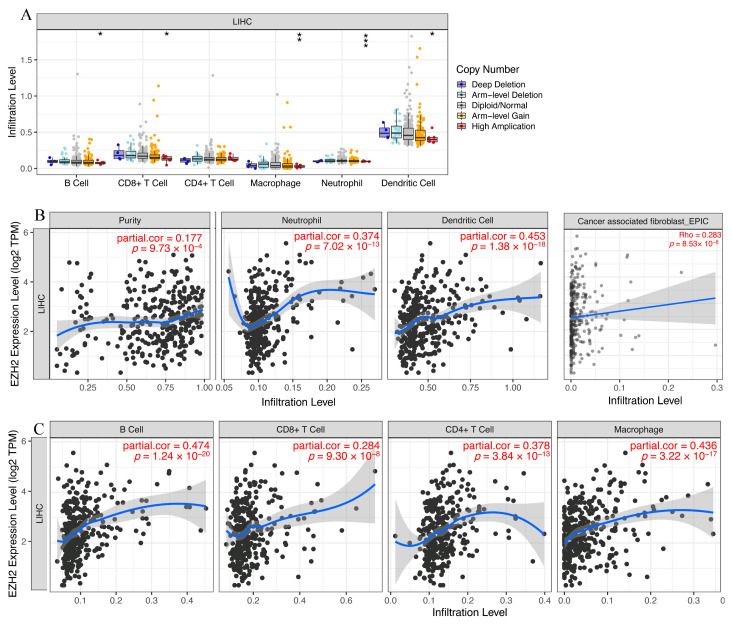
The relationship of immune cell infiltration with the *EZH2* level in HCC. (**A**) The infiltration level of various immune cells under different copy numbers of *EZH2* in HCC. (**B**,**C**) The correlation of the *EZH2* expression level with neutrophil, cancer-associated fibroblast (CAF), dendritic cell, B cell, CD8+ T cell, and macrophage infiltration levels in HCC.

**Figure 9 genes-13-00876-f009:**
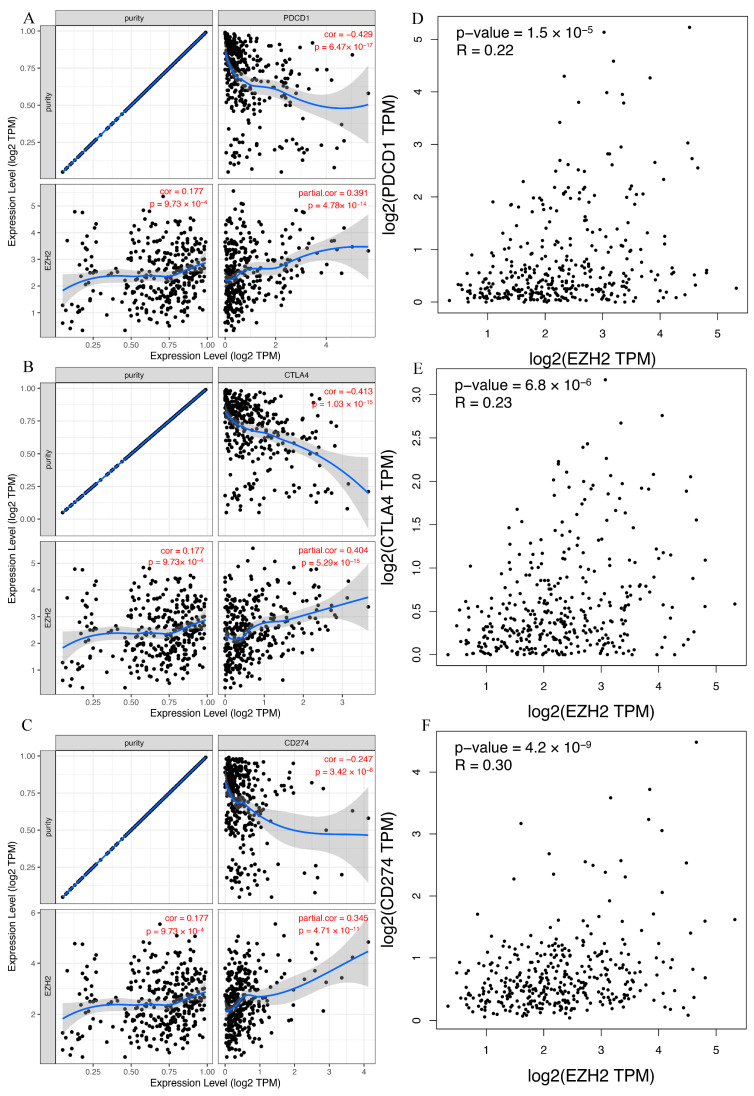
The relationship between EZH2 expression and immune checkpoint genes in HCC. (**A**,**D**) PDCD1, (**B**,**E**) CTLA4, and (**C**,**F**) CD274.

**Figure 10 genes-13-00876-f010:**
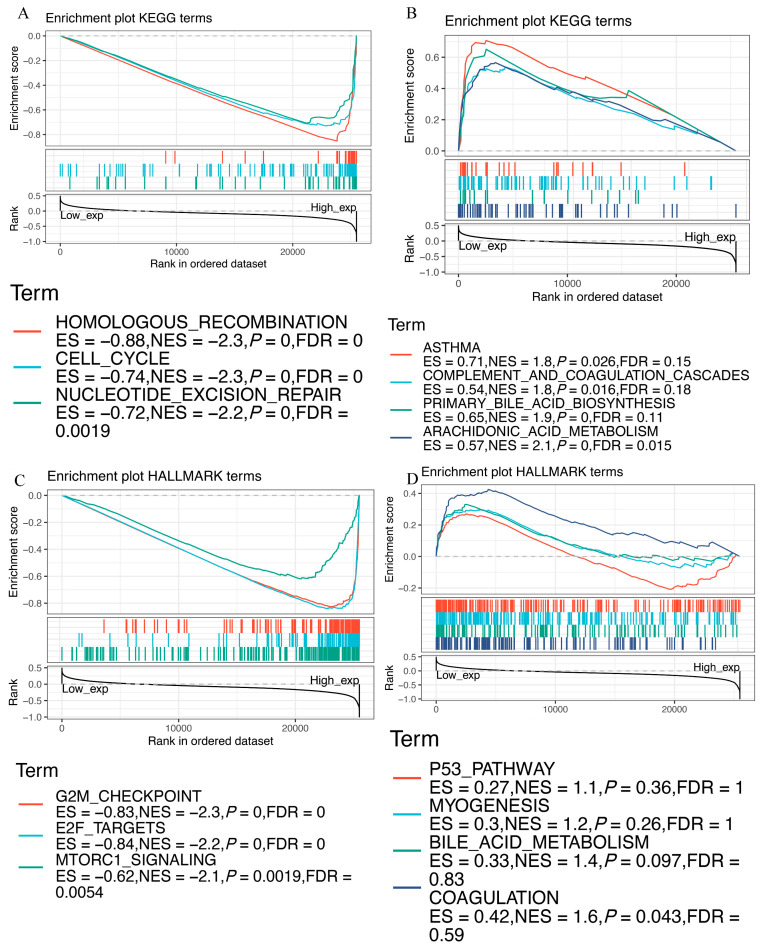
GSEA for samples with high and low *EZH2* expression. (**A**) Enriched gene sets in the KEGG collection by samples with high *EZH2* expression. (**B**) Enriched gene sets in KEGG by samples with low *EZH2* expression. (**C**) Enriched gene sets in the HALLMARK collection by samples with high *EZH2* expression. (**D**) Enriched gene sets in HALLMARK by samples with low *EZH2* expression.

**Figure 11 genes-13-00876-f011:**
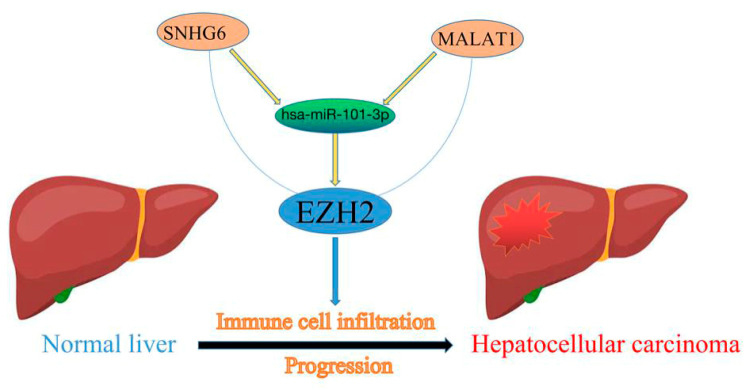
The model of the SNHG6/MALAT1-hsa-miR-101-3p-EZH2 axis in the carcinogenesis of HCC.

**Table 1 genes-13-00876-t001:** The correlation between EZH2 expression and chemotactic activity for immune cells.

Immune Cells	Chemokine	Cor	*p* Value
Monocytes/macrophages	CCL2	0.054	2.98 × 10^−1^
	CCL3	0.07	1.79 × 10^−1^
	CCL5	0.12	**2.06 × 10^−2^**
	CCL7	0.161	**1.86 × 10^−3^**
	CCL8	0.144	**5.54 × 10^−3^**
	CCL13	0.137	**8.24 × 10^−3^**
	CCL17	0.034	5.19 × 10^−1^
	CCL22	0.102	5.04 × 10^−2^
T lymphocytes	CCL2	0.054	2.98 × 10^−1^
	CCL1	0.166	**1.31 × 10^−3^**
	CCL22	0.102	5.04 × 10^−2^
	CCL17	0.034	5.19 × 10^−1^
Mast cells	CCR1	0.25	**1.06 × 10^−6^**
	CCR2	0.142	**6.20 × 10^−3^**
	CCR3	0.247	**1.48 × 10^−6^**
	CCR4	0.196	**1.42 × 10^−4^**
	CCR5	0.249	**1.20 × 10^−6^**
	CXCR2	0.133	**1.03 × 10^−2^**
	CXCR4	0.297	**5.58 × 10^−9^**
Eosinophils	CCL11	0.069	1.82 × 10^−1^
	CCL24	−0.006	9.07 × 10^−1^
	CCL26	0.283	**2.85 × 10^−8^**
	CCL5	0.12	**2.06 × 10^−2^**
	CCL7	0.161	**1.86 × 10^−3^**
	CCL13	0.137	**8.24 × 10^−3^**
	CCL3	0.07	1.79 × 10^−1^
Neutrophils	CXCL8	0.38	**6.12 × 10^−4^**

*EZH2*: Enhancer of Zeste Homolog 2; *p* values less than 0.05 are shown in bold.

**Table 2 genes-13-00876-t002:** Correlation analysis between *EZH2* and biomarkers of immune cells in HCC determined using the GEPIA and TIMER databases.

		GEPIA		TIMER	
		*R*	*p*	*R*	*p*
B cells	CD19	0.110	1.10 × 10^−1^	0.244	**1.92 × 10^−6^**
	CD79A	0.083	**7.90 × 10^−5^**	0.131	**1.18 × 10^−2^**
CD8+ T cells	CD8A	0.200	**1.40 × 10^−5^**	0.180	**5.05 × 10^−4^**
	CD8B	0.220	**7.70 × 10^−3^**	0.168	**1.17 × 10^−3^**
CD4+ T cells	CD4	0.140	8.20 × 10^−1^	0.217	**2.44 × 10^−5^**
M1 macrophages	NOS2	−0.012	**0.00 × 10^−0^**	0.003	9.48 × 10^−1^
	IRF5	0.440	5.00 × 10^−1^	0.482	**5.97 × 10^−23^**
	PTGS2	0.035	**2.80 × 10^−3^**	0.088	9.22 × 10^−2^
M2 macrophages	CD163	0.160	**2.40 × 10^−4^**	0.101	5.21 × 10^−2^
	VSIG4	0.190	**2.00 × 10^−3^**	0.100	5.41 × 10^−2^
	MS4A4A	0.160	8.30 × 10^−1^	0.102	5.07 × 10^−2^
Neutrophils	CEACAM8	0.011	**2.40 × 10^−11^**	0.094	6.95 × 10^−2^
	ITGAM	0.340	2.20 × 10^−1^	0.289	**1.44 × 10^−8^**
	CCR7	0.064	**2.40 × 10^−5^**	0.083	1.10 × 10^−1^
Dendritic cells	HLA-DPB1	0.220	9.00 × 10^−2^	0.154	**2.91 × 10^−3^**
	HLA-DQB1	0.088	**1.70 × 10^−5^**	0.146	**4.87 × 10^−3^**
	HLA-DRA	0.220	**1.20 × 10^−4^**	0.167	**1.22 × 10^−3^**
	HLA-DPA1	0.200	**7.40 × 10^−4^**	0.158	**2.28 × 10^−3^**
	CD1C	0.170	**6.50 × 10^−8^**	0.114	**2.85 × 10^−2^**
	NRP1	0.280	**1.10 × 10^−6^**	0.263	**2.67 × 10^−7^**
	ITGAX	0.250	1.10 × 10^−1^	0.348	**5.15 × 10^−12^**

*HCC*: hepatocellular carcinoma; *EZH2*: Enhancer of Zeste Homolog 2; *p* values less than 0.05 are shown in bold.

## Data Availability

The datasets used and analyzed in the present study are available from the corresponding authors on reasonable request.

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
