# Peer review of "High Expression of EZH2 Mediated by ncRNAs Correlates with Poor Prognosis and Tumor Immune Infiltration of Hepatocellular Carcinoma"

_genes, 2022, doi:10.3390/genes13050876_

Round 1
Reviewer 1 Report
The manuscript from Chen et al. discussed the potential role of EZH2 in hepatocellular carcinoma by analyzing public sequencing datasets. Then they built a ceRNA network of SNHG6 and MALAT1/ hsa-miR-101-3p/EZH2 regulatory pathways. They also find the relationship of immune cell infiltration with EZH2 level in HCC. However, without any mechanism of study, it is hard to convince readers to believe that the EZH2 ceRNA network in HCC. It is of the opinion of this reviewer that the authors should revise the manuscript accordingly.
1. The writing and organization of the manuscript is very difficult to read.
- Line 15, “...abnormal expressed EZH2 are…” should be changed to “...the abnormal gene expression of EZH2 is..”.
- Line 20-21, “and corresponding miRNAs” should be deleted.
- Line 22, “or” needed to be changed to “and”
- Line 24: What is the meaning of “target miRNA of EZH2”? Did the author want to say “miRNA targeting EZH2”?
- Line 25: What is the meaning of “…MALAT1 were determined to be…”?
- Line 26: “correlation analysis suggested/showed/revealed…”, the author shouldn’t use “correlation analysis found…”.
2. The authors should also experimentally validate the SNHG6 and MALAT1/ hsa-miR-101-3p/EZH2 regulatory network.
3. What is the base-pair information for hsa-miR-101-3p–EZH2, hsa-miR-101-3p–SNHG6 and hsa-miR-101-3p–MALAT1? The author should compare the minimum free energy (MFE) of above RNA duplex structures to further confirm the ceRNA network.
4. Line 140-141: how can the author summarize “EZH2 plays an important role in development and progression of human cancers “only based on localization analysis?
5. In Figure 2A, where is the EZH2 expression data of ACC, DLBC, LAML, LGG, MESO, OV, TCGT, SARC, THYM, UCS, UVM normal tissues?
6. In Figure 3B, DFS analysis shows that EZH2 is an unfavorable marker in BLCA, KIRP and LIHC. Why does the author only talk about BLCA and LIHC in the manuscript (line 174-175)? How about KIRP?
7. Please note the time unit for Figure 3C-M.
8. Please make sure the figures, legends and citations were correct for Figure 4C and 4D (Line 207 and 210).
9. What is the meaning of “upstream lncRNAs of hsa-miR-101-3p”? Are they potential target lncRNAs by hsa-miR-101-3p?
10. Line 92, Please note that the information on whether the statistical test used was one-sided or two-sided.
11. Please uniform the size of text in Figures.
Author Response
Dear editor and reviewers
Re: Manuscript ID: genes-1692883 and Title: High Expression of EZH2 Mediated by ncRNAs Correlates with Poor Prognosis and Tumor Immune Infiltration of Hepatocellular Carcinoma.
Thank you for your letter and for the reviewers' comments concerning our manuscript entitled “High Expression of EZH2 Mediated by ncRNAs Correlates with Poor Prognosis and Tumor Immune Infiltration of Hepatocellular Carcinoma.” (ID: genes-1692883). Those comments are all valuable and very helpful for revising and improving our paper, as well as the important guiding significance to our researches. We have studied comments carefully and have made correction which we hope meet with approval. The main corrections in the paper and the responds to the reviewers' comments are as flowing:
Responds to the reviewer’s comments:
Comments from reviewer 1:
The manuscript from Chen et al. discussed the potential role of EZH2 in hepatocellular carcinoma by analyzing public sequencing datasets. Then they built a ceRNA network of SNHG6 and MALAT1/ hsa-miR-101-3p/EZH2 regulatory pathways. They also find the relationship of immune cell infiltration with EZH2 level in HCC. However, without any mechanism of study, it is hard to convince readers to believe that the EZH2 ceRNA network in HCC. It is of the opinion of this reviewer that the authors should revise the manuscript accordingly.
- The writing and organization of the manuscript is very difficult to read.
- Line 15, “...abnormal expressed EZH2 are…” should be changed to “...the abnormal gene expression of EZH2 is..”.
- Line 20-21, “and corresponding miRNAs” should be deleted.
- Line 22, “or” needed to be changed to “and”
- Line 24: What is the meaning of “target miRNA of EZH2”? Did the author want to say “miRNA targeting EZH2”?
- Line 25: What is the meaning of “…MALAT1 were determined to be…”?
- Line 26: “correlation analysis suggested/showed/revealed…”, the author shouldn’t use “correlation analysis found…”.
Author’s response: Thank you for pointing out our issue. The whole text of the manuscript had been revised by a professional editing company (MDPI) with English native speaker for grammar and spelling check, and the editing certificate was also uploaded. If you have any questions, please do not hesitate to contact me.
Line 15-16, According to your opinion, we have changed “...abnormal expressed EZH2 are…” to “...the abnormal gene expression of EZH2 is..”
Line 20-21, According to your opinion, we have deleted “and corresponding miRNAs”.
Line 21-22, According to your opinion, we have changed “or” to “and”.
Line 25 The meaning of “target miRNA of EZH2” is miRNA targeting EZH2.
Line 24-25, The meaning of “SNHG6 and MALAT1 were determined to be directly targeted by hsa-miR-101-3p” is “SNHG6 and MALAT1 were identified as upstream lncRNAs of hsa-miR-101-3p.
Line 26, According to your opinion, we have changed “correlation analysis found…” to “correlation analysis revealed…”
- The authors should also experimentally validate the SNHG6 and MALAT1/ hsa-miR-101-3p/EZH2 regulatory network.
Author’s response: Considering that experimental verification takes a lot of time. It is very difficult to complete reagent purchase, sample collection and experimental supplement within 10 days. Therefore, this is also an aspect that we need to work hard to improve in the future. For experimental verification, our research group has been actively preparing for qRT-PCR analysis based on cell and tissue levels to verify the expression of EZH2, hsa-miR-101-3p, SNHG6 and MALAT1 in cancer and adjacent tissues. Due to the recent epidemic of COVID-19 in Hangzhou, China, the reagents purchased cannot be delivered and the school laboratory is closed, we regret that we are unable to conduct experimental verification. At the same time, TCGA is recognized as a database with high credibility and can be used for tumor research, so we think that the research based on TCGA is credible. All in all, the reviewers, editor-in-chief, and members of our research group have a common purpose to make this study better for journal publication. Finally, I sincerely thank the reviewers and editors for their careful and prudent working attitude, and the questions raised are very constructive.
- What is the base-pair information for hsa-miR-101-3p–EZH2, hsa-miR-101-3p–SNHG6and hsa-miR-101-3p–MALAT1? The author should compare the minimum free energy (MFE) of above RNA duplex structures to further confirm the ceRNA network.
Author's response: Thank you for your valuable comments. the base-pair information for hsa-miR-101-3p–EZH2 (Figure 5G, Line 219-220), hsa-miR-101-3p–SNHG6 (Figure 6D, Line 236-238) and hsa-miR-101-3p–MALAT1(Figure 6E, Line 236-238) were discussed. These results further validate the ceRNA network. Due to limited conditions, we cannot compare the minimum free energy (MFE) of the above RNA double-stranded structures. Therefore, this is also an aspect that we need to work hard to improve in the future.
- Line 140-141: how can the author summarize “EZH2 plays an important role in development and progression of human cancers “only based on localization analysis?
Author's response: Thank you for the reminder, we carefully reviewed the literature and checked the content of our paper, and finally decided to delete “EZH2 plays an important role in development and progression of human cancers”. Thank the reviewers again for their careful review, which makes our paper more rigorous.
- In Figure 2A, where is the EZH2 expression data of ACC, DLBC, LAML, LGG, MESO, OV, TCGT, SARC, THYM, UCS, UVM normal tissues?
Author's response: Thanks for your question. In this study, the EZH2 expression data of ACC, DLBC, LAML, LGG, MESO, OV, TCGT, SARC, THYM, UCS, UVM normal tissues were from The Cancer Genome Atlas (TCGA) and Genotype-Tissue Expression (GTEx) database, and elaborated in the Materials and methods section (Line 96-98).
- In Figure 3B, DFS analysis shows that EZH2 is an unfavorable marker in BLCA, KIRP and LIHC. Why does the author only talk about BLCA and LIHC in the manuscript (line 174-175)? How about KIRP?
Author's response: We feel sorry for our carelessness, we have corrected these errors in the revised manuscripts (Line 179-180).
- Please note the time unit for Figure 3C-M.
Author's response: Thank you for pointing out our issue. We have added the time unit for Figure 3A-K.
- Please make sure the figures, legends and citations were correct for Figure 4C and 4D (Line 207 and 210).
Author's response: We feel sorry for our carelessness, we have corrected these errors in the revised manuscripts (Line 214-219).
- What is the meaning of “upstream lncRNAs of hsa-miR-101-3p”? Are they potential target lncRNAs by hsa-miR-101-3p?
Author's response: Yes, the meaning of “upstream lncRNAs of hsa-miR-101-3p” is potential target lncRNAs by hsa-miR-101-3p.
- Line 92, Please note that the information on whether the statistical test used was one-sided or two-sided.
Author's response: Thank you for the reminder, after our research, the statistical test used was two-sided. We corrected it in the manuscript (Line 98). Thank the reviewers again for their careful review, which makes our paper more rigorous.
- Please uniform the size of text in Figures.
Author's response: Thank you for the reminder, we unified the size of text in Figures to make it readable.
We tried our best to improve the manuscript and made some changes in the manuscript. These changes will not influence the content and framework of the paper. And marked in red in revised paper. We deeply appreciate your consideration of our manuscript, and we look forward to receiving comments from the editor and reviewers. If you have any queries, please don’t hesitate to contact me at the address below.
Thank you and best regards.
Yours sincerely.
Corresponding author: Qiyong Li, Shusen Zheng
E-mail: zjliqiyong@163.com, shusenzheng@zju.edu.cn

Reviewer 2 Report
In this paper, the authors here are focusing on non-coding RNA that regulates EZH2 and they chose to use The Cancer Genome Atlas (TCGA) and The Genotype-Tissue Expression (GTEx) data to perform differential expression analysis and prognostic analysis. As well as considering Encyclopedia of RNA Interactomes (ENCORI) database to predict candidate miRNAs and lncRNAs that may bind to EZH2 and corresponding miRNAs. They performed expression analysis, correlation analysis or survival analysis, and their results showed upregulation of EZH2 in the majority of tumor types.
They concluded that SNHG6 and MALAT1/ hsa-miR-101-3p/EZH2 axis are potential regulatory pathways in the progression of HCC.
Overall the paper has good scientific content, some parts can be improved and of course to make sure that the paper follows the IJMS style and the spelling to be carefully checked.
- In Abstract Line 21, you mentioned “... the comprehensive analysis (including expression analysis, correlation analysis or survival analysis) ...” but why do you say “or survival analysis” and not and. It is not clear if you performed them all or which one?
- The Introduction part is a bit too short and more information would be required. For example line 53: “In recent years, an increasing number of studies have investigated that EZH2 may be a novel molecule involved in HCC progression, as well as a potential prognostic biomarker and therapeutic target.” - after “an increasing number of studies”, please give examples and cite as you mention what has been published, and how is that relevant. Not just add the refs at the end as it is confusing where those citations should belong, after which statement.
- Figure 1. A and D - Increase the text size as is not clear to read. C. The text should be moved a bit more down to be clearer. E - to be consistent with the spelling. Either all with capital letters (EZH2) or all with small capital letters
- Figure 2. B - Please increase the text size to be readable
- Figure 3. I would encourage you to split this figure into 2-3 figures and properly explain each. Right now the figure is too crowded and difficult to follow. I would do one figure containing the current parts A, B, - B underneath A. Then the “Disease-specific survival” results are set into another figure, rearranging them maximum 3 graphs per row, and the same for the other ones. Each of them needs more explanation and interpretation.
- I recommend rounding the P-values in the main text as well as in table B from Figure 4, the values with 3 units after the comma. As well the legend of the colors used to highlight is not clear.
- The same as previously for Figure 4. I Recommend rearranging it to be understandable. The text in C, D, E, and F is not readable.
- Line 218 - be consistent with spelling (eg Upstream - upstream)
- Figure 6 - Remove the blue borders of each representation, let the white background, and increase the text size. As well I suggest removing the gridlines. More explanations and interpretations of the figures are needed.
- Table 1. The same as previously for P-value and be consistent, or 2 or 3 numbers after the comma for all
- The same in Figure 8. The P-value number
- Figure 9 - increase the text size
- In the Discussion section describe more about your own results and interpretation of your own results. Relate the literature citations with your own work.
Author Response
Dear editor and reviewers
Re: Manuscript ID: genes-1692883 and Title: High Expression of EZH2 Mediated by ncRNAs Correlates with Poor Prognosis and Tumor Immune Infiltration of Hepatocellular Carcinoma.
Thank you for your letter and for the reviewers' comments concerning our manuscript entitled “High Expression of EZH2 Mediated by ncRNAs Correlates with Poor Prognosis and Tumor Immune Infiltration of Hepatocellular Carcinoma.” (ID: genes-1692883). Those comments are all valuable and very helpful for revising and improving our paper, as well as the important guiding significance to our researches. We have studied comments carefully and have made correction which we hope meet with approval. The main corrections in the paper and the responds to the reviewers' comments are as flowing:
Responds to the reviewer’s comments:
Comments from reviewer 2:
In this paper, the authors here are focusing on non-coding RNA that regulates EZH2 and they chose to use The Cancer Genome Atlas (TCGA) and The Genotype-Tissue Expression (GTEx) data to perform differential expression analysis and prognostic analysis. As well as considering Encyclopedia of RNA Interactomes (ENCORI) database to predict candidate miRNAs and lncRNAs that may bind to EZH2 and corresponding miRNAs. They performed expression analysis, correlation analysis or survival analysis, and their results showed upregulation of EZH2 in the majority of tumor types.They concluded that SNHG6 and MALAT1/ hsa-miR-101-3p/EZH2 axis are potential regulatory pathways in the progression of HCC. Overall the paper has good scientific content, some parts can be improved and of course to make sure that the paper follows the IJMS style and the spelling to be carefully checked.
- In Abstract Line 21, you mentioned “... the comprehensive analysis (including expression analysis, correlation analysis or survival analysis) ...” but why do you say “or survival analysis” and not and. It is not clear if you performed them all or which one?
Author’s response: Thank you for pointing out our issue. The whole text of the manuscript had been revised by a professional editing company (MDPI) with English native speaker for grammar and spelling check, and the editing certificate was also uploaded. If you have any questions, please do not hesitate to contact me.
Line 21, According to your opinion, we have changed “or” to “and”.
- The Introduction part is a bit too short and more information would be required. For example, line 53: “In recent years, an increasing number of studies have investigated that EZH2 may be a novel molecule involved in HCC progression, as well as a potential prognostic biomarker and therapeutic target.” - after “an increasing number of studies”, please give examples and cite as you mention what has been published, and how is that relevant. Not just add the refs at the end as it is confusing where those citations should belong, after which statement.
Author’s response: Thank you for your valuable comments, to increase the credibility of EZH2 as a therapeutic target in hepatocellular carcinoma. We reviewed the literature and added in the introduction. For example, (Line 57-61) Liu et,al.[1] have suggested that EZH2/miR-622/CXCR4 as a potential adverse prognostic factor and therapeutic target for HCC patients. Similarly Bae et,al. shared the same view that overexpression of EZH2 was an independent biomarker for poor outcomes of HCC, and EZH2 may be used as a therapeutic target in patients with HCC[2].
Reference
- Liu H, Liu Y, Liu W, Zhang W, Xu J. EZH2-mediated loss of miR-622 determines CXCR4 activation in hepatocellular carcinoma. Nat Commun 2015; 6:8494.
- Bae AN, Jung SJ, Lee JH, Lee H, Park SG. Clinical Value of EZH2 in Hepatocellular Carcinoma and Its Potential for Target Therapy. Medicina (Kaunas) 2022; 58(2).
- Figure 1. A and D - Increase the text size as is not clear to read. C. The text should be moved a bit more down to be clearer. E - to be consistent with the spelling. Either all with capital letters (EZH2) or all with small capital letters.
Author's response: Thank you for your careful reminder. we increased the size of text in Figures to make it readable (Figure 1A and 1D). We moved the text down to make the Figure clearer to read (Figure 1C). We have unified the spelling format of EZH2 (Figure 1E). Thank the reviewers again for their careful review, which makes our paper more rigorous.
- Figure 2. B - Please increase the text size to be readable.
Author's response: Thank you for your careful reminder. After cautious organization and modification of Figure 2, all text is easily legible without magnification. (Figure 2B).
- Figure 3. I would encourage you to split this figure into 2-3 figures and properly explain each. Right now the figure is too crowded and difficult to follow. I would do one figure containing the current parts A, B, - B underneath A. Then the “Disease-specific survival” results are set into another figure, rearranging them maximum 3 graphs per row, and the same for the other ones. Each of them needs more explanation and interpretation.
Author's response: Thank you for your constructive suggestions. According to your opinion, we divided Figure 3 into two Figures (Figure 3 and Figure 4).
- I recommend rounding the P-values in the main text as well as in table B from Figure 4, the values with 3 units after the comma. As well the legend of the colors used to highlight is not clear.
Author's response: Thank you for your careful reminder. We have unified the writing format of p values with 3 units after the comma. We reduced the highlight of the image color, all text is easily legible without magnification (Figure 5).
- The same as previously for Figure 4. I Recommend rearranging it to be understandable. The text in C, D, E, and F is not readable.
Author's response: Thank you for your careful reminder. After cautious organization and modification of Figure 5, all text is easily legible without magnification.
- Line 218 - be consistent with spelling (eg Upstream - upstream)
Author's response: Thank you for the reminder, we have unified the spelling format of Upstream. Thank the reviewers again for their careful review, which makes our paper more rigorous.
- Figure 6 - Remove the blue borders of each representation, let the white background, and increase the text size. As well I suggest removing the gridlines. More explanations and interpretations of the figures are needed.
Author's response: Thank you for your careful reminder. After cautious organization and modification of Figure 7, all text is easily legible without magnification.
(1). We have removed the blue border of each Figure with white background.
(2). we increased the size of text in Figures to make it readable.
- Table 1. The same as previously for P-value and be consistent, or 2 or 3 numbers after the comma for all.
Author's response: Thank you for your careful reminder. We have unified the writing format of p values with 2 units after the comma.
- The same in Figure 8. The P-value number
Author's response: Thank you for your careful reminder. We have unified the writing format of p values with 2 units after the comma.
- Figure 9 - increase the text size
Author's response: Thank you for your careful reminder. After cautious organization and modification of Figure 10, all text is easily legible without magnification.
- In the Discussion section describe more about your own results and interpretation of your own results. Relate the literature citations with your own work.
Author's response: Thank you for your constructive suggestions. We have described more about our results and interpretations in the discussion section (Line 363-365, Line 373-377, Line 380-383 ).
We tried our best to improve the manuscript and made some changes in the manuscript. These changes will not influence the content and framework of the paper. And marked in red in revised paper. We deeply appreciate your consideration of our manuscript, and we look forward to receiving comments from the editor and reviewers. If you have any queries, please don’t hesitate to contact me at the address below.
Thank you and best regards.
Yours sincerely.
Corresponding author: Qiyong Li, Shusen Zheng
E-mail: zjliqiyong@163.com, shusenzheng@zju.edu.cn

Round 2
Reviewer 1 Report
In this manuscript, Chen et al. built a ceRNA network of SNHG6 and MALAT1/ hsa-miR-101-3p/EZH2 regulatory pathways. The manuscript is much improved after the revision. I just have one comment on this work: the Fig4, Fig6, Fig9 and Fig10 were not readable in the revised manuscript. The authors must be very cautious in dealing with the final version.